

# The origin and role of biological rock crusts in rocky desert weathering

Nimrod Wieler[1], Hanan Ginat[2], Osnat Gillor[1]*# and Roey Angel[3]*#

[1]Zuckerberg Institute for Water Research, Blaustein Institutes for Desert Research, Ben Gurion University of the Negev Sede
Boqer Campus, Israel

[2]The Dead Sea and Arava Science Center, Israel

[3]Soil and Water Research Infrastructure and Institute of Soil Biology, Biology Centre CAS, Czechia

* These authors contributed equally to this study

Correspondence to: Roey Angel[3] (roey.angel@bc.cas.cz) or Osnat Gillor[1] (gilloro@bgu.ac.il)

**Abstract**

In drylands, microbes that colonise rock surfaces were linked to erosion because water scarcity excludes traditional
weathering mechanisms. We studied the origin and role of rock biofilms in geomorphic processes of hard lime and dolomitic
rocks that feature comparable weathering morphologies though originating from arid and hyperarid environments,
respectively. We hypothesised that weathering patterns are fashioned by salt erosion and mediated by the rock biofilms that
originate from the adjacent soil and dust. We used a combination of microbial and geological techniques to characterise
rocks morphologies and the origin and diversity of their biofilm. Amplicon sequencing of the SSU rRNA gene suggested
that bacterial diversity is low and dominated by Proteobacteria and Actinobacteria. These phyla formed laminar biofilms
only on rock surfaces that were exposed to the atmosphere and burrowed up to 6 mm beneath the surface, protected by
sedimentary deposits. Unexpectedly, the microbial composition of the biofilms differed between the two rock types and was
also distinct from the communities identified in the adjacent soil and settled dust, showing a habitat-specific filtering effect.
Moreover, the rock bacterial communities were shown to secrete extracellular polymeric substances that form an evaporation
barrier, reducing water loss rates by 65-75%. The reduced water transport rates through the rock also limit salt transport and
its crystallisation in surface pores, which is thought to be the main force for weathering. Concomitantly, the biofilm layer
stabilises the rock surface via coating and protects the weathered front. Our hypothesis contradicts common models, which
typically consider biofilms as weathering-promoting agents. In contrast, we propose the microbial colonisation of mineral
surfaces acts to mitigate geomorphic processes in hot, arid environments.

## 1 Introduction

In arid and hyperarid stony deserts, bedrock surfaces are typically barren and free of vegetation or continuous soil mantle.
When these surfaces are exposed to atmospheric conditions, they undergo weathering processes that shape the landscape
(Smith, 2009). Weathering is an in-situ set of processes that include physical, chemical and mechanical forces that result in
the breakdown and transport of the shuttered fragments from the parent rock. Weathering can appear in a range of sizes and
morphologies (Smith et al., 2005), including gravel shattering (Amit et al., 1996), surface crazing (Smith, 1988), ventifacts
(Smith, 1988), microrills (Smith, 1988; Sweeting and Lancaster, 1982) and cavernous patterns [also known as tafoni,
honeycomb or pitting (Mustoe, 1983; Viles, 2005). Weathering is an essential, though often neglected, element in the overall
denudation of hot deserts.
Cavernous weathering is one of the most frequently occurring weathering patterns that have been observed in various
regions across the globe, including humid and arid, cold and hot, coastal and inland sites (Bruthans et al., 2018). In the





Negev Desert, Israel, cavernous weathering patterns are common in carbonate rocks in arid and hyperarid regions. Upon exposure to the atmosphere, these rocks develop a carbonate coating, termed calcrete or dolocrete (respective to limestone or dolomite) by displacive and replacive cementation of calcium or dolomite onto the rock surface (Wright and Wacey, 2004; Alonso-Zarza and Wright, 2010). Following the cementation processes, typical honeycomb features are formed on the

exposed parent rock, typified by pits separated by thin walls that are coated by the calcrete or dolocrete. Recent studies suggest that microbial activity also promotes the processes of calcrete and dolocrete formation (Alonso-Zarza et al., 2016; Alonso-Zarza and Wright, 2010).

The accepted conceptual model for the formation of cavernous rock weathering in hot deserts involves the presence of permeable rocks that are subjected to soluble salts and repeated episodes of drying-rewetting cycles (Goudie et al., 2002;

Smith, 1988; Smith et al., 2005). The proposed mechanisms assume that cavernous weathering results from physicochemical processes including salt crystallization (Cooke, 1979; Scherer, 2004), incipient fractures (Amit et al., 1996), exfoliation (Shtober-Zisu et al., 2017), or stress-erosion (Bruthans et al., 2014; McArdle and Anderson, 2001). Recently, Bruthans and colleagues (2018) conclusively demonstrated the superiority of the hydraulic hypothesis (moisture flux followed by salt crystallisation at the boundary layer) over case hardening model, in a temperate climate.

In addition, biological mechanisms have been proposed to promote rock weathering through mechanisms such as flaking via colony growth (Viles, 2012), acidification by bacterial extractions (Garcia-Pichel, 2006; Warscheid and Braams, 2000) or alkalization during photosynthesis by cyanobacteria (Büdel et al., 2004). In contrast, it was proposed that micro- and macro-organisms colonisation can mitigate weathering in temperate, coastal regions (McIlroy de la Rosa et al., 2014; Mustoe, 2010) through encrustation or protection from direct rain impact. Yet, it is not clear which of these mechanisms dominates or what

is the relative contribution of chemical vs biological processes to weathering in arid environments.

Microorganisms colonising rocks form a hardy biofilm known as the biological rock crusts (BRC), which is common in most arid and hyperarid regions worldwide (Gorbushina, 2007; Lebre et al., 2017; Pointing and Belnap, 2012). Epilithic communities colonising rock surfaces are ubiquitous in arid environments, while hyperarid rocks, which experience increased radiation and desiccation, are dominated by endolithic communities that colonise internal rock pores

(Makhalanyane et al., 2013; Pointing and Belnap, 2012; Viles, 1995). The BRC communities include cyanobacteria and other phototrophic bacteria and heterotrophic bacteria, but very low abundances of archaea, fungi or algae (Lang-Yona et al., 2018). However, the BRC inoculum was not resolved and was proposed to originate from settled dust (Viles, 2008), or the surrounding soil (Makhalanyane et al., 2015).

The goal of this study was to illuminate the origin and role of BRCs in cavernous weathering of exposed limestone and

dolomite rocks in arid and hyperarid regions. We predicted that the BRC communities on exposed rock surfaces will resemble either the ever-present dust or the surrounding soil, supporting a subset of adapted taxa from both sources. We further hypothesised that the cavernous weathering morphologies of exposed rocks result from salt mobilisation by dew, causing crystallisation pressure under atmospheric conditions. The developed rock biofilms clog the surface rock pores through secretion of extracellular polymeric substances (EPS), lowering evaporation and slowing the salt crystallisation, but

also stabilising the exfoliated rocks preventing further weathering. Thus, the presence of a BRC mitigates the geomorphic processes. To test our hypotheses, we applied a holistic approach combining field observations, geological, geotechnical and molecular microbiology characterisation elucidating BRCs' morphology, origin and role in arid cavernous weathering.

## 2 Materials and Methods

### 2.1 Study site

We focused on two sites in the Negev Desert, Israel: Sede-Boqer – an arid site and Uvda Valley – a hyperarid site (Fig. S1, Table 1). Both sites are rocky terrains underlined predominantly by carbonate rock slopes consisting of limestone, dolomite,





chalk, marl, clay and chert from the Cretaceous to Eocene age. Our analyses compared samples from the limey Turonian age Shivta Formation located in the arid region with samples from the dolomitic Turonian age Gerofit Formation located in the hyperarid environment. The Negev Desert, Israel, maintains arid to hyperarid conditions since the Holocene and has an aridity index (P/PET) of 0.05-0.005 (Amit et al., 2010; Bruins, 2012) similar to other arid and hyperarid areas worldwide,

e.g., the Namib and Atacama Deserts (Azua-Bustos et al., 2012; Viles and Goudie, 2007). The long-term aridity of the Negev Desert makes it a reliable site for testing the cross-influence between BRCs and geological substrates.

### 2.2 Field sampling

Twenty-four rock samples were collected along rocky slopes facing northward, comprising: twelve limestone samples from the limey Turonian age Shivta Formation at the arid site (30.88N34.78E, WGS 84 Grid; samples named: SB 1-12) and

twelve dolomite samples from limey-dolomitic Turonian age Gerofit Formation at the hyperarid site (29.94N34.97E, WGS 84 Grid; samples named: UV 1-12) during November and December, 2014. Concomitantly, six soil samples (ca. 500 g each) were collected, half from the arid (named: SBSoil 1-3) and a half from the hyperarid (named: UVSoil 1-12) sites. Each rock or soil sample is a composite of four sub-samples that were pooled and homogenised in the lab.

We also collected settled dust samples using glass beads traps (Goossens and Rajot, 2008). The traps were placed on

December 2013 and collected three months later in the arid (samples named: SBDust 1-2) and hyperarid (samples named: UVDust 1-2) sites. Each dust sample was a composite of two sub-samples that were pooled and homogenised in the lab.

### 2.3 Geological analyses

The geological methods used in this study are based on direct field observations and detailed characterisation of the subjected lithologies (i.e., Limestone and Dolomite) which included morphology (thin sections), mineral components [X-ray

powder diffraction (XRD)], porosity and permeability (Automatic Gas Permeameter Porosimeter), and elastic properties (Schmidt hammer): Petrographic thin sections, 30 μm thick, were prepared for each lithology to test the main components in both the BRC and host rocks examined under a light microscope (Zeiss, Oberkochen, Germany). XRD analysis of mineral components (Sandler et al., 2015) was conducted on the BRC and host rocks using three replicates each. Powdered samples were scanned using X'Pert[3] Powder diffractometer equipped with a PIXcel detector (Panalytical Malvern, Almelo,

Netherlands). Scanning range was: 3 – 70° 2θ, step size 0.013°, speed 70.1 s per step.  Total effective porosity (ϕ) and permeability (k) tests (Scherer, 1999) were performed using Automatic Gas Permeameter Porosimeter (Core Laboratories, Houston, Texas, USA) on twelve rock core cylinder samples, with 18.5 mm radius and 26.5 mm height. Six samples were taken from each lithology, each set of six samples were prepared in two orthogonal directions providing the normal to bedding and parallel to bedding. Before testing porosity and permeability, samples were oven dried at a temperature of 110°c

for 24 h. Schmidt hammer (Lassen, Aarhus, Denmark) tests were applied in the field (Goudie, 2016; Viles et al., 2011). Twenty measurements were carried out for each lithology.

### 2.4 FTIR and stable isotope analysis

Fourier transform infrared spectroscopy (FTIR) analysis was conducted for testing the presence of extracellular polymeric substances (EPS) on the rock surfaces while the host rock was used for comparison. The spectra were recorded using a

Vertex 70 FTIR spectrometer (Bruker, Billerica, MA, USA) with a 4 cm$^{-1}$ scan resolution. One to two mg of pulverised rock was taken from each sample (n = 2), and the spectra were measured twice collected over a wavenumber range 4000-600 cm$^{-1}$, and a baseline correction was carried out. The spectral absorption bands, indicative for EPS, were identified according to published information (Ferrando et al., 2018).



For δ $^{13}$C and δ $^{18}$O analysis, 1-2 mg of rock surface powder (i.e., calcite or dolomite) was obtained using a Microdrill (Dremel, Racine, WI, USA) along with a cross-section of the rock crust and its host rock. Four profiles measurements of δ $^{13}$C and δ $^{18}$O were performed on samples UVSL 5-6 from the hyperarid site and NWSH 1-2 from the arid site. Measurements (in duplicate) of δ $^{18}$O-H$_2$O and δ $^{13}$C-DIC were performed on gas source isotope ratio mass spectrometer (GS-

IRMS; Thermo Fisher Scientific, Waltham, MA, USA) coupled to a Gas Bench II interface (Thermo) after CO$_2$ equilibration or CO$_2$ extraction by acidification for δ $^{18}$O-H$_2$O and δ $^{13}$C-DIC, respectively. The samples were calibrated against internal laboratory standards: Vienna Standard Mean Ocean Water (VSMOW) and carbonate standard NBS19. δ $^{13}$C values were also referenced against VSMOW and valued for carbonate relative to Vienna PeeDee Belemnite (VPDB) standard as previously described (Uemura et al., 2016) with SD of 0.1‰. All values are reported in per-mil (‰).

**2.5 Desiccation experiment**

To test the effect of biological rock crusts on water transport rates in the rock, clogging and desiccation experiments were performed on sixteen rock core cylinders from both lithologies (limestone and dolomite). Rock cylinders (∅ 37 mm, 6.5 cm) were drilled using a rock core drill. Each set of eight rock cores from the two different lithologies included four rock cores that were kept intact, and four rock cores that their BRC was mechanically removed using a diamond saw (Dremel, Racine,

WI, USA) to a depth of 5 cm. Each cylinder was immersed in distilled water for 72 h, covered with epoxy (Devcon) and aluminium foil, leaving only the upper base of the crusted and bare cylinders uncovered to allow evaporation. The cylinders were then weighed (t=0), incubated in an oven dried at a temperature of 44 °C for 48 h and weighed every 2 h during the first 12 h and then every 6 h to determine the residual water content. Second-degree polynomial functions were fitted using function stats::lm to determine evaporation rates and compared using ANOVA in R.

**2.6 DNA Extraction PCR amplification and sequencing**

For DNA extraction from all rocks, the surface (ca. 100 cm$^2$) was scraped using a rasp (66-67 HRC hardness; Dieter Schmid, Berlin, Germany) that was cleaned with 70% technical-grade ethanol before each sampling. DNA was then extracted from 0.5 g of the homogenised sample using bead-beating in the presence of a CTAB buffer and phenol, according to a previously published extraction protocol (Angel et al., 2012). A 466-bp fragment of the *16S rRNA* gene was amplified using the

universal bacterial primers 341F (CCTAYGGGRBGCASCAG) and 806R (GGACTACNNGGGTATCTAAT) flanking the V3 and V4 region (Klindworth et al., 2012). Library construction and sequencing were performed at a DNA Services Facility (University of Illinois at Chicago, USA) using a MiSeq sequencer (Illumina, San Diego, CA, USA) in the 2 × 250 cycle configuration (V2 regent kit). The raw sequencing data were deposited into the EMBL-ENA SRA database (https://www.ebi.ac.uk/ena/) and can be found under study accession PRJNA381483.

**2.7 Sequence processing and analysis of bacterial communities**

Paired reads generated by the MiSeq platform were quality filtered and clustered into OTUs using the UPARSE pipeline (Edgar, 2013), with modifications. Contig assembly was done using the fastq_mergepairs command. Then, contigs were dereplicated with the derep_fulllength command, and singleton sequences were removed. OTU centroids were then determined with the cluster_otus command (set at 3% radius). Abundances of OTUs were determined by mapping the

filtered contigs (before dereplication, including singletons) to the OTU centroids using the usearch_global command (set at 0.97% identity). Following these steps, a total of ca. 1.4 G reads remained. OTU representatives were classified using mothur's implementation of a Naïve Bayesian sequence classifier (Schloss et al., 2009; Wang et al., 2007) against the SILVA 119 SSU NR99 database (Quast *et al.*, 2013). All downstream analyses were performed in R V3.4.4 (R Core Team, 2016). Data handling and manipulation were done using package phyloseq (McMurdie and Holmes, 2013). For alpha-

diversity analysis, all samples were subsampled (rarefied) to the minimum sample size using bootstrap subsampling at 1000





iterations, to account for library size differences, while for beta-diversity analysis library size normalisation was done using GMPR (Chen et al., 2018). The ACE richness estimate (O'Hara, 2005) and Shannon's H diversity index were calculated using function EstimateR in the vegan package (Oksanen et al., 2018) and tested using ANOVA and Tukey HSD in the stats package. Variance partitioning and testing were done using PERMANOVA (McArdle and Anderson, 2001) function

vegan::adonis using Horn-Morisita distances. Differences in phyla composition between the sample types were tested using the non-parametric Scheirer Ray Hare Test (Mangiafico, 2018); function rcompanion::scheirerRayHare) followed by the *post hoc* Mann-Whitney Test (function stats::wilcox.test) and FDR corrected using the Benjamini-Hochberg method (Ferreira and Zwinderman, 2006); function stats::p.adjust). Detection of differentially abundant OTUs was done using ALDEx2 (Fernandes et al., 2014). Plots were generated using packages ggplot2 (Wickham, 2016) and ggtern (Hamilton, 2017).

**3 Results and discussion**

**3.1 Field and mineralogical observations**

Weathering features were observed in about 30% of exposed rocks sampled from both arid and hyperarid sites. Neither the prevalence of weathering nor its morphology seemed to differ between sites despite the different climates and underlying geology. In all cases, weathering type was classified as tafoni or honeycomb weathering (Goudie et al., 1997; Groom et al.,

2015); Fig. 1A), and it was coupled with the presence of sub-aerial biofilm, burrowed underneath the surface and protected by sedimentary deposits (Fig. 1B). The weathering and presence of the crusts were restricted to the atmospherically exposed parts of the rock. The presence of identical weathering morphology and prevalence in different climates and lithologies challenges the current model, which assumes that surface permeability, moisture and the presence of salts as primary factors control weathering rates (Goudie et al., 2002; Smith, 1988; Smith et al., 2005). However, the lack of correlation between

tafoni weathering magnitude and climate has already been reported (Brandmeier et al., 2011).
To study the possible differences between these sites, we performed geological characterisation of 10 limestone and dolomite rocks collected from the arid and hyperarid sites, respectively, testing for mineral content, porosity, permeability and elasticity. As expected, our results showed different lithological parameters between the limestone and dolomite rocks (Table 1), yet they displayed similar weathering features. Moreover, petrographic thin section analysis showed that on both

rock types, crusts had developed to a similar thickness of 1-6 mm, irrespective of climatic conditions including mean annual precipitation (Fig. 1C). However, microclimatic conditions, like dew or surface temperature may impact local morphologies. Also, the thin sections showed that the crusts are composed of masses of micritic to microsparitic minerals that form laminated structure (Fig. 1C). Such laminated structures indicate that the crusts are stage four terrestrial calcretes and dolocretes, suggesting a mature crust phase (Alonso-Zarza and Wright, 2010). The calcretes and dolocretes identified on the

rocks' surface reject previously suggested impact of mineralised networks or case hardening (McBride and Picard, 2004). In fact, the detection of mature calcretes could serve as an indication of atmospheric exposure but was also suggested to result from biogenic activity (Alonso-Zarza and Wright, 2010; Goudie, 1996).

**3.2 Composition and chemical characteristics of the rock crusts**

To test our hypothesis that the crusts are biogenic and involved in rock weathering processes, we characterised their origin

and nature. An XRD analysis of the crust layers and bedrocks showed that the crusts are composed of similar mineralogy as their respective host rocks, indicating that local weathering, rather than dust deposition, is the source of crust generation (Table 1).
The biogenic nature of the crusts was confirmed using a cross-section analysis of the stable carbon and oxygen isotopes ratios in the crust and host rock (Fig. 2A). For both limestone and dolomite, values of $\delta^{13}C$ increased between the crust and

the host rock layers and ranged from -4.1‰ in the calcrete to -0.9‰ in the limestone bed, and from 0.2‰ in the dolocrete to

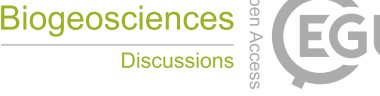



2.0‰ in the dolomite bed. Such values are typical indicators to carbon isotope exchange of primary marine $CaCO_3$ (abundant in the bedrock) with $CO_2$ released by microbial respiration (i.e. of carbon originating from photosynthesis) with subsequent precipitation of pedogenic calcrete (Brlek and Glumac, 2014; Mora et al., 1991). Analysing $\delta^{13}C$, together with $\delta^{18}O$, compositions of pedogenic carbonates is a useful way of reconstructing paleo-vegetation (e.g., C3/C4 plant ratio

(Ehleringer et al., 1997; Mora et al., 1991). Our $\delta^{13}C$ results go along with $\delta^{13}C$ values collected from speleothems (secondary mineral deposits formed in caves) collected in the central and southern Negev Desert (Vaks et al., 2010) that were also dated to end of the Pliocene (the past 2.5 million years). The low ratio detected here (Fig 2A) and by Vaks et al. (2010) suggest that the Negev region has been able to support only limited vegetation for at least 2.5 Ga, if so then the role of the crust in shaping the morphology of the rock surfaces was considerable.  These results support the hypothesis that

calcretes and dolocretes are of biogenic origin, and therefore the crust can be referred to as BRC. Moreover, they indicate similar developmental trajectory for both the calcretes and dolocretes that is independent of aridity or lithological parameters.

In contrast, the trend in values of $\delta^{18}O$ differed between rock types. In the limestone rocks, the ratio ranged from -3.0‰ to -6.8‰ between the BRC and the host rock, while in the dolomite it was higher, ranging from -5.4‰ to -0.6‰. The decrease

in $\delta^{18}O$ in the host limestone rock could be explained by meteoric water substitution (Sandler, 2006). In contrast, the more negative $\delta^{18}O$ values in the dolocrete compared to the host dolomite are attributed to isotopic differentiation of meteoric water due to condensation (Rayleigh distillation) and could result from the large distance from the Mediterranean Sea (that is the primary source of rainfall in the area) compared to closer limestone rocks. In speleothems, similar patterns in $\delta^{18}O$ values were reported in the central and southern Negev Desert (Vaks et al., 2010). The results suggest that the calcrete and

dolocrete studied here have been experiencing arid to hyperarid climates since the Pleistocene, alluding to the possible source of rain. A similar study conducted in the Thar Desert in India also inferred sedimentary rocks stable isotope patterns to paleoclimate (Andrews et al., 1998).

To study the potential role of the BRC in the weathering process, its composition was characterised using FTIR as was previously reported (Sheng et al., 2010). We focused on the functional groups and element compositions in EPS or microbial

aggregates and found a distinct peak in the BRC layers ranging between 1020-1040 cm$^{-1}$ in both limestone and dolomite rocks that was absent from the host rocks (Fig. 2B). This peak is indicative of the presence of EPS from bacterial origin (Shirshova et al., 2006), pointing to the significant components of asymmetric and symmetric stretching of $PO_2^-$ and $P(OH)_2$ in phosphate as well as vibrations of C-OH and C-C bonds found in polysaccharides and alcohols (Jiang et al., n.d.). These results provide a strong support for the biogenic nature of the crust, since EPS is a common feature of many if not most

biofilms (Drews et al., 2006). The detected EPS could serve several functions in BRC such as dust-particle trap to collect the dust and its nutrients, a binding agent to individual members of the biofilm (Davey and O'toole, 2000), or a protective agent by decreasing evaporation and retaining moisture and shielding from radiation (Or et al., 2007; Roberson and Firestone, 1992).

Based on these findings, we hypothesised that BRC could in fact act as a mitigator during the weathering process by

clogging the pores on the surface of the rock and thereby minimising capillary rise. Consequently, crystallisation of dissolved salts, considered to be the primary mechanism for rock weathering, is mitigated. To test this hypothesis, we performed a desiccation experiment to estimate water loss from the rock surfaces covered with BRC. The results suggest that both in limestone and dolomite rocks water moves through the rock and is lost to evaporation two or three times faster in the absence of BRC than when it is present (Fig. 3). Considering that salt transport due to hydraulic movement is a dominant

weathering mechanism (Huinink et al., 2004), reduced evaporation due to BRC coverage will also inevitably lead to decrease weathering rate. Moreover, the obtained results stand in contrast to similar measurements performed on temperate sandy stones that showed no significant effect of BRC on water transport rates (Slavík et al., 2017).

### 3.3 The microbial composition and origin of the BRCs





To elucidate the identity of the bacterial communities on the limestone and dolomite BRCs, we applied a multiplexed barcoded amplicon sequencing of the small subunit RNA gene (SSU rRNA). In addition, we compared the BRC communities to those of samples of the surrounding soil and settled dust in order to deduce the origin for the rock biofilm. As expected, we found poor and low-diversity of the BRC communities. The communities of the BRC showed an average of

182 and 129 observed, 354 and 315 predicted phylotypes, and Shannon's H was 3.8 and 3.3 (Fig. 4A; Table S2), for arid limestone and hyperarid dolomite, respectively, with no significant difference between the rock types.

The surrounding soil was significantly richer and more diverse ($P < 0.05$) in the arid site (416 and 746 observed and predicted OTUs and Shannon's H = 5.6 on average), and equally rich but slightly more diverse in the hyperarid site (221 and 466 observed and predicted OTUs and Shannon's H = 3.8, on average). The diversity of the dust samples were was as poor as

the BRC's (169 and 107 observed and predicted OTUs and Shannon's H = 3.0 and 1.5, on average) and did not differ between sites (Fig. 4A; Table S2). The number of observed OTUs in the soil and their diversity scores were somewhat lower in this study compared to reports from similar environments (Barberán et al., 2014; Lang-Yona et al., 2018; Šťovíček et al., 2017) however these could be due to sequencing technologies and depth. The lower richness and diversity in hyperarid vs arid samples and the BRC and dust vs. soil samples is expected and comparable with trends reported in other works (Angel

and Conrad, 2013; Barberán et al., 2014; Lang-Yona et al., 2018).

Beta-diversity analysis showed statistically significant differences between samples on the OTU-level by climate, sample type, and to a small extent also their interaction, using variance partitioning. These parameters were found to significantly contribute to the differences in bacterial communities accounting for 22%, 40% and 3.8% of the variance (Fig. 4B, Table S3). Pairwise comparisons further showed that the two BRCs significantly differed from one another ($P < 0.01$) and also

from their surrounding soil and dust samples ($P < 0.05$ in all cases; Table S3). The bacterial community in the samples was typical for drylands, mostly dominated by members of the phyla Proteobacteria, and Actinobacteria followed by Deinococcus–Thermus, Chloroflexi, Bacteroidetes, Cyanobacteria, Acidobacteria, Firmicutes, and Gemmatimonadetes (Fig 4C, Table S4). Similar communities have repeatedly been reported for arid and hyperarid soils and rocks (Angel and Conrad, 2013; Barberán et al., 2014; Lang-Yona et al., 2018). While cyanobacteria are typically the main primary producers in the

soil and rock communities (Weber et al., 2016), recent studies showed that other autotrophs may also contribute significantly to the energy balance of these biofilms (Ji et al., 2017).

The BRCs of the two rock types differed in the relative abundance and composition of major phyla. Most notably, Proteobacteria were significantly more dominant in the hyperarid compared to the arid samples ($P = 0.02$) comprising on average 21% and 44% of the community in the limestone and dolomite BRC, respectively. In contrast, the Actinobacteria showed an opposite trend ($P = 0.03$) comprising on average 42% and 21% of the community in the limestone and dolomite

BRC, respectively. The two BRCs also differed in their composition of Firmicutes, Gemmatimonadetes and Chloroflexi ($P < 0.03$; Fig 4C, Table S4).

The soil samples generally showed similar trends on the gross taxonomic level as their respective BRC samples. While none of the phyla differed significantly between the hyperarid BRC and the soil, the phyla Deinococcus–Thermus, Acidobacteria,

Firmicutes, and Gemmatimonadetes significantly differed between limestone BRC and the surrounding arid soil ($P < 0.04$; Table S4). Lastly, the arid and hyperarid dust samples were dominated by members of the Proteobacteria, with other phyla comprising only a minor fraction of the community (with a notable exception of Bacteroidetes that dominated one of the dust samples). However, these differences were not significant, probably due to the small sample size (Table S4).

Despite the general similarities in community composition between samples on the phylum level, many of the OTUs found

in each sample were unique to the BRC, soil or dust as evident by the ternary diagrams (Fig. 4D). Direct analysis of the differences in the OTUs detected 130 (10%) differentially abundant OTUs in the dolomite BRC and 74 (6%) differentially abundant OTUs in the limestone BRC (Fig. S2). Similarly, several differentially abundant OTUs were also detected when comparing the BRCs to their respective soil and dust samples. However, these differentially abundant OTUs were fewer, probably due to the small dust sample size (Fig. S2).





The BRC bacterial communities were previously described (Kuhlman et al., 2006; Lang-Yona et al., 2018; Wong et al., 2010a, 2010b) but their origin and role in geomorphological processes were not considered. Our results suggest that despite the similarity in morphology and magnitude of rock weathering features in the arid limestone and hyperarid dolomites, the two BRCs harboured distinct microbial communities, differing in over 16% of the OTUs and their composition at the

phylum level. Moreover, despite the spatial proximity and continuous interaction between the limestone and dolomite surface to their respective surrounding soils and dust particles, the bacterial communities of the BRCs were distinct. The abilities of bacteria to disperse, settle and persist in a given location could be an important factor resulting in the biogeographic patterns observed here. The difference between arid and hyperarid soil communities could result from the local contribution of aeolian material that might affect the loess soil diversity (Crouvi et al., 2008). Alternatively, the

hyperarid site experience slow pedological processes while arid soil formation was enhanced (Amit et al., 2011) resulting in disparate bacterial communities. The three matrices (BRC, soil and settled dust) studied here sparsely shared their bacterial communities and specifically, the BRC community had little in common with the soil or dust communities (Fig 4). This demonstrates the ecological filtering effect of the rock surfaces, which imposes unique abiotic challenges on the microbes living on it (Horner-Devine & Bohannan, 2006). This also suggests that the BRCs cannot be regarded as passive deposits of

microbial cells originating from the surrounding soil or dust, but rather it is a specific subset of adapted microbes that can persist and form a biofilm under these unique conditions.

### 3.4 The role of BRC in arid rock weathering - synthesis

Honeycomb weathering patterns are prevalent worldwide and are found in both humid and dry ecosystems. According to contemporary models, this form of weathering is the result of the transport of dissolved salts through the rock and their

eventual crystallisation in surface pores, leading to fractures and eventual flaking of rock material (Rodriguez-Navarro et al., 1999). In this study, we found that weathering patterns and magnitude are similar on rocks from both arid and hyperarid sites, despite the differences in precipitation and lithologies. In arid and hyperarid regions, BRCs were shown to form once the rock is exposed to the atmosphere (Pointing and Belnap, 2012). A developed crust of biological origin was microscopically and isotopically apparent on all weathered rocks and was shown to be supported by EPS (Fig. 2). Similar to

weathering magnitude, the BRCs showed no observable differences in form or depth despite the different aridity and lithology. Both BRCs comprised bacterial taxa that are typical for xeric environments (Pointing and Belnap, 2012) and included many heterotrophs but also dominant phototrophs or otherwise autotrophic members (Fig. 4). The two BRCs did differ in their bacterial communities at the OTU and higher taxonomical levels, demonstrating a discrepancy between composition and function. The BRC communities also differed from their surrounding soil and dust, indicative of the

specialism of the colonising taxa to rock environment. In the absence of mineralized networks or case hardening (i.e., addition of cementing agent to rock matrix material) we conclude that calcrete and dolocrete were formed through the colonisation of microorganisms and the secretion of EPS, serving as a thin biofilm (Brantley et al., 2011; Weber et al., 2016). Our results further suggest that this biogenic layer mitigates evaporation and reduces water transport, hence alleviating salt crystallisation pressure in the rock pours (Scherer, 2004). Crystallization of calcium sulphate and sodium chloride solutions,

which are abundant in these soils, was shown to build pressure within pores and stress rocks (Scherer, 2004; Sperling and Cooke, 1985). This process is enhanced under low relative humidity and rapid evaporation and compromises the durability of the rocks (Rodriguez-Navarro et al., 2003; Rodriguez-Navarro and Doehne, 1999). Our results suggest that the presence of BRC decreases evaporation rates (Fig 3) and thus attenuate the crystallisation pressure and reduces damage to the rocks. Moreover, the BRC may also stabilise the rock following exfoliation preserving the weathered structure.

Arid weathering features, which lead to debris formation result from a dynamic balance between the erosive salt forces and the mitigating effects of the BRC. The role of microbial biofilms in the protection of surfaces from mineral weathering was extensively studied for biomineralisation and sedimentation processes (Adams et al., 1992; Dupraz et al., 2009). Yet, the role of BRC in weathering processes under atmospheric conditions in the desert has not been considered before. We propose that




microbial colonisation of mineral surfaces protects the rocks from weathering by mitigating salt crystallisation and stabilising the weathered front. Rock weathering processes are typically believed to be controlled at different scales ranging from the climatic scale, down to local conditions at the site and eventually the microscale (Smith, 2009; Sperling and Cooke, 1985; Viles, 2001). The results presented here suggest that in arid environments, microscale conditions determine the

magnitude of weathering that shape the landscape.

**Author contributions**

N.W., H.G., O.G. and R.A. conceptualised the study; N.W. performed the field and lab work; N.W. and R.A. analysed the data; N.W., H.G., O.G. and R.A. wrote the manuscript

**Competing interests**

The authors declare no conflict of interests

**Acknowledgments**

RA was supported by BC CAS, ISB & SoWa RI (MEYS;  projects LM2015075, EF16_013/0001782 – SoWa Ecosystems Research).

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



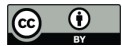

**Table 1: Geological parameters of the subjected lithologies.**

| Rock properties | Limestone (arid; Shivta Formation) | | | Dolomite (hyperarid; Gerofit Formation) | | |
|---|---|---|---|---|---|---|
| | Dolomite | Calcium | Quartz | Dolomite | Calcium | Quartz |
| BRC mineralogy (%) | 0 | 95 | 3 | 90 | 2 | 1 |
| Host rock mineralogy (%) | 0 | 95 | 0 | 95 | 0 | 1 |
| Porosity (%) | $13.5 \pm 2.2$ | | | $8.25 \pm 1.3$ | | |
| Permeability (miliDarcy) | $0.1 - 3.8$ | | | $0.05 - 0.41$ | | |
| Surface penetration resistance (kg cm$^{-1}$) | $100 - 365$ | | | $130 - 230$ | | |



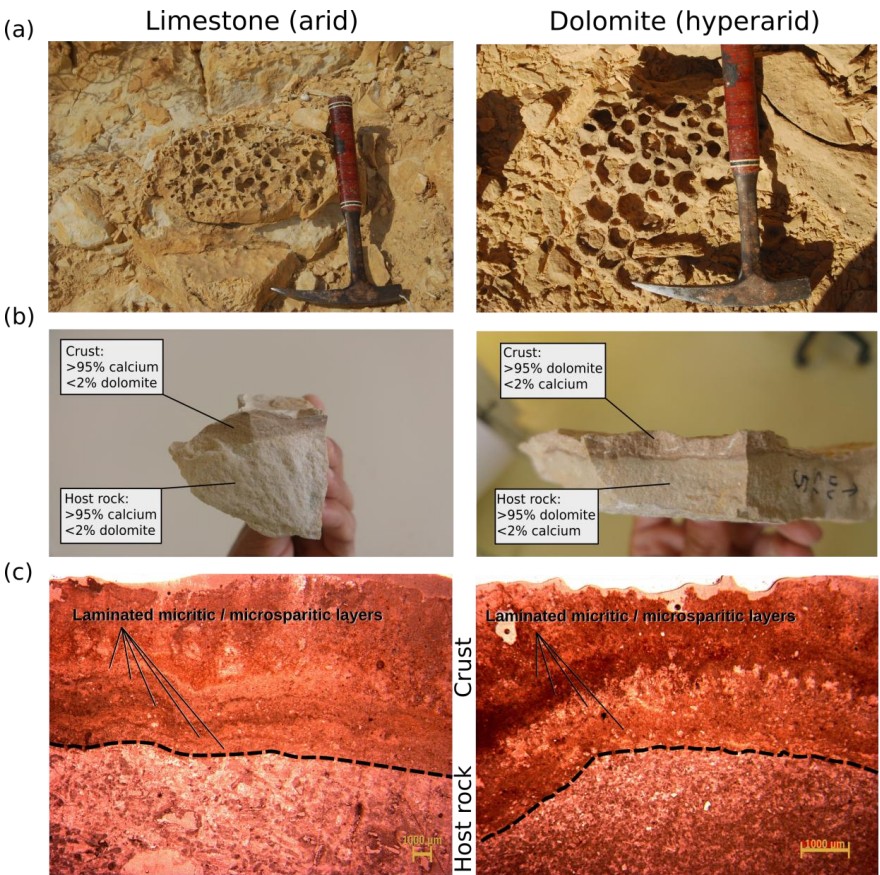

**Figure 1: (a)** Comparable weathering features in the exposed limestone and dolomite rocks on both sites as noted in field outcrops (hammer for scale, 30 cm long). **(b)** Visual presence of a rock crust with similar thickness (3-6 mm) in both rock types. The crust's mineralogical composition matched that of the host rock. **(c)** Thin-section analysis of the rocks showing lamination structure in the BRCs. Dashed lines indicate the interface between BRC host rocks. BRC's mineralogy includes micritic to microsparitic dolomitic or calcitic crystals.





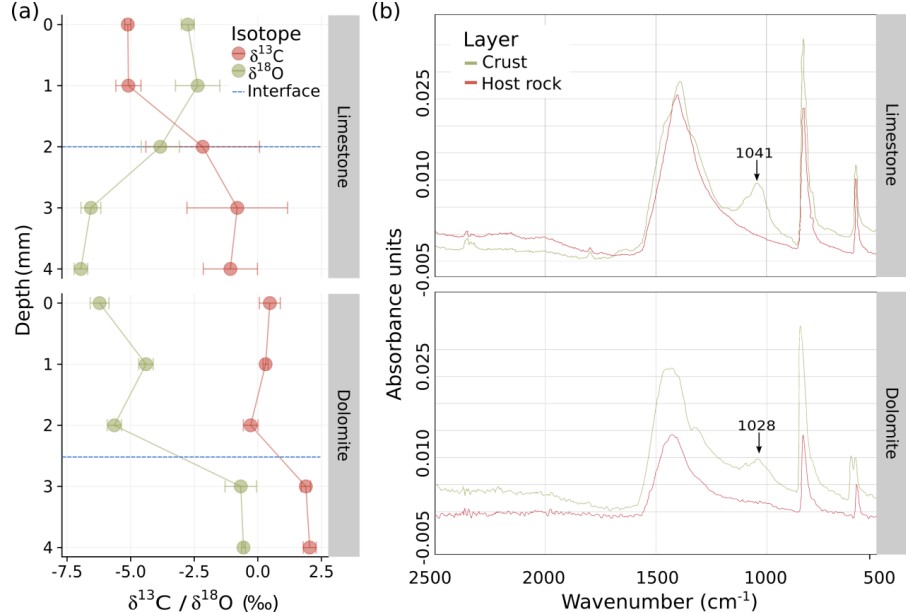

**Figure 2: (a)** Fourier Transform Infrared (FTIR) analysis of limestone (top) and dolomite (bottom) BRCs indicating the presence of extracellular polymeric substances (EPS) molecules through the distinctive peak ranging between 1020-1040 cm$^{-1}$, which was absent from the host rocks. **(b)** Carbon and oxygen isotope-ratio depth profiles of the limestone (top) and

5      dolomite (bottom) BRC's in comparison to their host rocks.





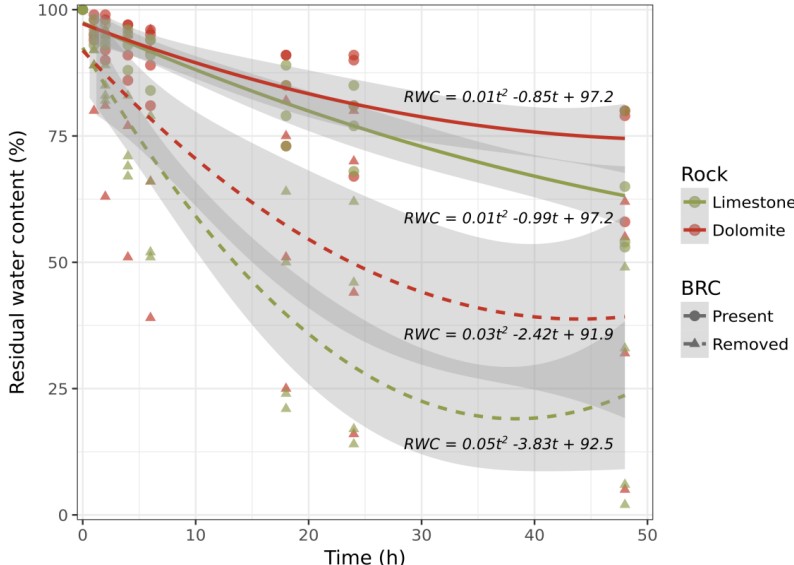

**Figure 3:** Desiccation of rock cores in the presence and absence of BRC as a function of time, following full hydration. The curves indicate a second-degree polynomial line fitting (all fitted curves were statistically significant from each other in ANOVA tests with P values $< 0.01$ and $R^2 > 0.95$).



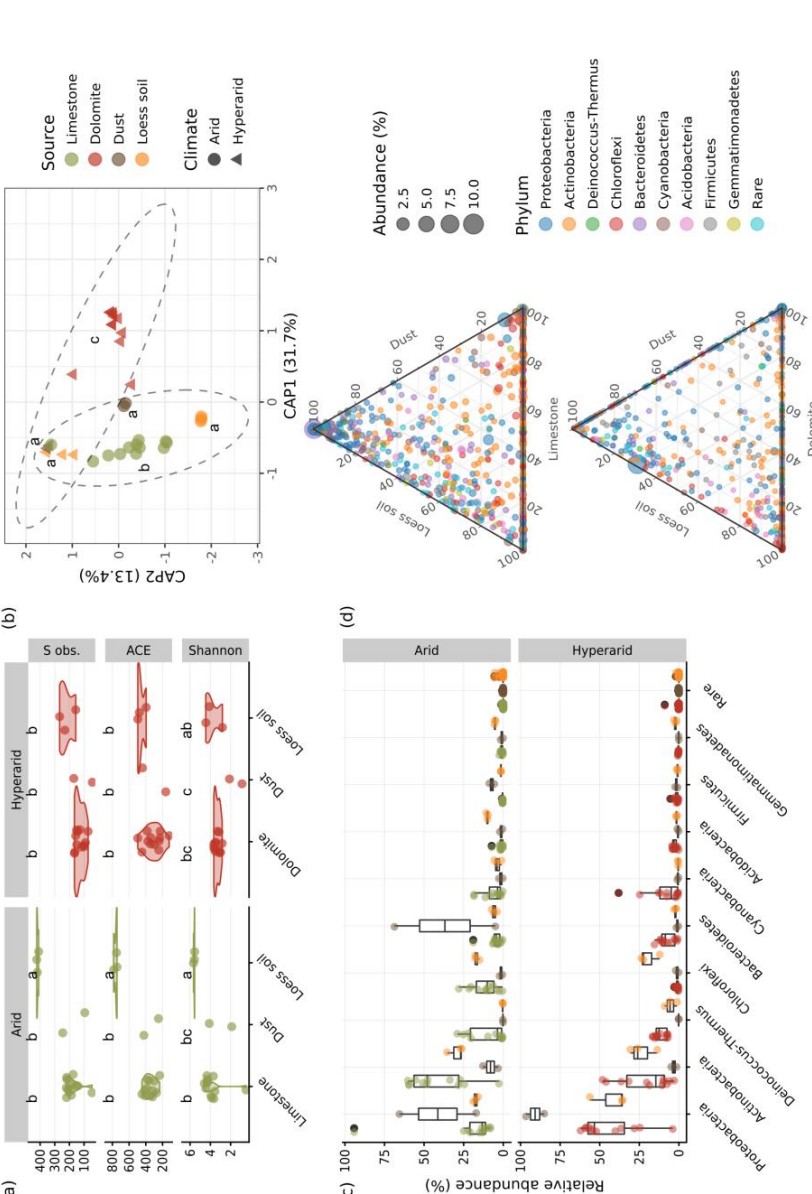

**Figure 4:** Microbial community features of the BRCs, the surrounding soils and settled dust in the two studied sites. **(a)** Comparison of the richness in the form of observed no. of OTUs (S obs.) and the predicted number of OTUs (ACE index), and a comparison of α-diversity (Shannon's H Index) between the different sample types. Identical lower-case letters indicate no statistical difference between groups in a Tukey's HSD test. **(b)** Clustering of sample types using a PCoA ordination based on Horn-Morisita distance matrix. Identical lower-case letters indicate no statistical difference between groups in a pairwise PERMANOVA test. Ellipses denote 95% confidence intervals around the arid and hyperarid samples assuming multivariate normal distribution. **(c)** Composition of bacterial phyla in the different sample types (see Table S4 for results of statistical tests in the relative abundance of different phyla between sample types). **(d)** Relative contribution of each bacterial OTU to the community composition of each sample type. Top – arid site, bottom – hyperarid site (see Figure S2 for statistical detection of preferentially abundant OTUs between each sample-type pair.

