# Peer review of "The origin and role of biological rock crusts in rocky desert weathering"

_Biogeosciences, 2018_

## Referee Comment (RC1) · Anonymous Referee #1 · 29 Nov 2018

General comments The manuscript: " The origin and role of biological rock crust in rocky desert weathering" by Wieler et al. is focused on the origin and role of rock biofilms in cavernous weathering in arid and hyperarid climate. Authors use multiple techniques to reveal the origin of biocrust and its effect on evaporation rate and thus weathering. Manuscript contains valuable information and its worth of publication. The answer to questions 1-15 in review instruction is positive, except the critical comments mentioned below. Please take into account that I am not expert on DNA techniques nor on statistical processing of such data, so I can not reliably review chapters 2.6, 2.7 and 3.3 from biological point of view in required depth. These chapters seems to be however clear and makes sense to person from other scientific branch.

Specific comments (P1 L12 means 1 page 12th line) It is unclear which portions of

honeycombs or tafoni surfaces were sampled for biogenic rock crust (BRC). Was is the outer surfaces or hollows (cavities)? From P2 L5 it seems that just outer surfaces are covered by BRC, but it is not clearly stated. It should be spelled our more clearly if BRC is missing in caverns or if it covers whole surface of tafoni. P1 L32 reference at the end of sentence is needed P2 L26 Fungi and algae are reported as common constituent of BRC by Slavik et al (2017)-cited in document P3 L1 there should be few more sentences given on characterization of limestone and dolomite: sedimentation settings, diagenesis, lithology, whether these rocks act as aquifer or aquitard, into which degree the water from rain infiltrates to them vs. surface runoff dominates P3 L4 rather than P/PET 0.05-0.005 you should write this ratio for both studied localities respectively (to show the difference between them). This ratio is in one of supplementary tables, but it should be also directly in the text. P3 L10 these samples were taken from 1) narrow walls of tafoni, 2) hollows of tafoni, 3) outer surfaces, which are not covered by tafoni or 4) inner material below tafoni hollows? This should be clear. Similarly for each method used is important which of these four types of material you used P3 L29 you mention measuring of porosity in direction normal and perpendicular to bedding. This is good idea, but please report also results from both measurements (table 1). Currently the direction is not distinguished there. Measured samples were without crust, with crust or crust itself? It should be more clearly spelled out in this (and also other) method(s), whether the underlying rock or crust was tested! If crust was not measured it will be valuable to measure the crust as well and compare it to underlying rock. P3 L30 It is generally recommended to do about 20 readings by Schmidt hammer per single obtained value. Your 20 measurements per lithology means 20 readings (1 value) or 20 sites measured each by ?20? readings? Please specify. Also in further text you use "elasticity" (P5 L23), "surface penetration resistance"(Table 1). This cannot be measured by Schmidt hammer, but it could be possibly derived by some formula. Did you measure it by other device? (Please characterize the device) or did you calculated that from rebound value of Schmidt hammer (then please provide the formula and reference). The terms "penetration resistance" elasticity of material should be unified in text

and tables, if they describe the same measured parameter. P4 L4and6 really it was delta18O in H2O? I would expect O in CO2 gas not H2O vapor. The sample is carbonate not water. Please check. P4 L7-8 Really SMOW was used? All values seems to be referenced to PDB standard to me (Fig. 2). Please check. P4 L17 What do you mean by "incubated"? bacteria were introduced to rock? P5 L16 "crusts were restricted to atmospherically exposed. . ." please change "atmospherically exposed" to more proper description. Do you mean that crust was missing in bottom of hollows? Or deeper below surface? This should be more clear. P5 L17 vs L19 Statement is not consistent. In first sentence you speak about weathering MORPHOLOGY in second you speak about weathering RATES. So if morphology is the same, this does not necessarily mean their rates are the same as well. I am afraid that this if fact not challenges the model. P5 L29-32. Text is unclear, please rewrite. P6 L35 "clogging the pores on the surface of the rock and thereby minimizing capillary rise". This statement is confusing. In fact the smaller the pores, the higher the capillary rise. The reason why biota affects capillary water is not the diminishing the size of pores but the presence hydrophobic organic matter. So please mention rather hydrophobicity here as explanation. P6 L42 Slavik et al 2017 reports DECREASE of hydraulic conductivity and capillary water absorption by 15-300 times and 2-33 times, respectively. So cited statement that BRC does NOT affects water transport rates is wrong. Only diffusion of water vapor was not effected by crust based on Slavik et al 2017. But in your case it could be the same situation: lowered evaporation is not necessarily due to low diffusion of vapor (only few if any well designed studies indicated that biocrust is capable even to affect vapor transport). Far more probably the decreased evaporation rate (which you observe on crust) is due to fact that capillary front is pushed below surface due to hydrophobic organic matter and thus diffusion occurs via more thick dry surface layer in case of BRC sample compare to bare rock core (longer diffusion path means far lower evaporation rate). Until the vapor diffusion is measured via BRC and bare rock and both rates are compared on your samples (e.g. by wet cup technique) it is impossible to say if evaporation rate is lowered by (i) lower diffusion rate or (ii) due to different geometry of capillary front. This

will be very valuable to test. P7 L13 From which depth the samples were taken? P8 L28and29 Sentence not clear. "…composition and function." of what? P9 L4 please specify which "microscale conditions"

Technical correction (P1 L12 means 1 page 12th line) P1 L13 replace "hard lime" by "limestone" P3 L12 if UVSoil had 3 samples, there should not be "UVSoil 1-12" but rather "1-3" P4 L10 Use rather "Evaporation experiment" then "Desiccation experiment" P5 L12 replace "weathering features" by "Cavernous weathering features" to be clear which weathering features you mean. P6 L8 please replace 2.5 Ga by 2.5 Ma P6 L14 there should be rather "In" then "between". The sentence is not much clear (it is unclear if values are concerning BRC, underlying rock, or both; but clearly not the boundary between them - Fig 2). Fig. 2 please add PDB standard to horizontal axis of fig. a Fig. 3 please replace "desiccation" by "drying"

---

## Referee Comment (RC2) · Anonymous Referee #2 · 4 Jan 2019

In the manuscript " The origin and role of biological rock crust in rocky desert weathering" by Wieler et al., the authors set out to characterize the microbial communities associated with rock crusts on limestone and dolomite host rocks sampled from arid regions. In this aspect they have succeeded. The authors also claim to have discovered how crust-associated microbial communities influence the mediation of weathering processes associated with these clasts. With respect to this second claim, the authors have only shown here that EPS associated with the microbial communities helps the rock surface to retain water, not that the water retention mitigates the weathering process via slowing crystal growth (as is claimed). The finding of EPS retaining water has been shown before in other environments (refs below), but in those studies, the retention of water was proposed to enhance weathering via various mechanisms, not

retard it. Perhaps if experiments showed that rock weathering decreased under EPS-free portions of the rock surface, I would find this second claim convincing, but these experiments/data are not present. I suggest that the authors rework the manuscript to focus only on the characterization of the community, and not on biogenicity aspects of the crust formation that are not supported by the research findings. In addition, I have the following concerns:

- The authors frequently misspell words that should contain the letter "z" but instead are spelled with an "s" (stabilise vs stabilize; colonise vs. colonize etc.). Perhaps this is a US vs British spelling difference, but the journal editors may want to clarify which style they want used.

- Page 2, Line 14, This sentence could be rewritten for clarity.

- Page 5 Line 12, please provide the number of samples that contain weathering features.

- Page 5 Line 38, I disagree with the authors' use of the terms "biogenic" to refer to the rock crust. Let's assume that the 13C depleted values results from the liberation of carbon from photosynthetic materials via respiration (there are other ways to get 13C-depleted carbonate, but let's just assume the mechanism the authors invoke is correct), that $CO_2$ should then be creating an acidic environment that does not necessarily favor carbonate formation. More importantly, a carbon contribution from respiration recorded in a carbonate does not make a rock crust any more "biogenic" than any carbonate that forms in any environment in which $CO_2$ is sourced from respiration, which could be any environment! I strongly recommend that the authors remove biogenic from these paragraphs, as the carbonate carbon isotope data do not demonstrate that living processes were necessary (or even important) for the carbonate crust formation.

- Page 6, Line 28, please provide the full citation information for the Jiang paper.

- Page 6, Line 29, the spatial correlation of a biofilm with a mineral precipitate DOES

NOT establish that the biofilm was involved in the formation of the mineral formation. As an example, the modern day La Brea tar pits contain abundant bacteria and archaea, that does not mean that those bacteria and archaea are responsible in any way for the presence of the tar of the fossils embedded in the tar, despite their spatial correlation. The same is true for our teeth, or for certain modern stromatolites. I'm certainly not saying that there aren't many cases where microbes are involved in mineral precipitation, there clearly are many, included microbes involved in carbonate formation, but in cases like this, it can be difficult to demonstrate this relationship and we should be careful with our words and our claims.

- Page7, Line 5, the description of the observed vs. predicted phylotypes (predicted by Chao/ACE) is unclear and should be better described.

- Page 8, "specialism" should be "specialization"

- Page 9, Line 9 says "were was" , but should be just "was"

- Page 9, Line 17 – this sentence could be rewritten for clarity

- Page 9, Line 18 – This sentence should probably be ended with "respectively)" to indicate which percentages with each parameter, or better yet rewrite the entire paragraph and give a sentence to each parameter.

- Figure 4b – the dust samples are hard to differentiate using the current color.

- I understand that the rock crusts studied here are not the same as the manganese oxidize-rich rock varnish that has been extensively studied elsewhere, but are the microbial communities similar or different? Would this be worth mentioning as a point of comparison? Some readers will be more familiar with those features.

- The authors propose that the microbial community should be similar to that of the surrounding soil, or incoming dust, if those are the sources, but then demonstrate with their amplicon results, that the communities on the rocks are substantially different from those in the soil and dust. This is an interesting result and worthy of publication for its

own sake in my view. I think the authors do a nice job with this part of the paper and should be commended.

- The presented results do appear to show that the biofilm contributes to the retention of water at the rock surface. However, this is not a new claim and there are numerous other papers in the older literature that also show this (Potts, M. (1999) Mechanisms of desiccation tolerance in cyanobacteria. Eur J Phycol 34: 319–328.; Decho, A.W. (2000) Exopolymer microdomains as a structuring agent for heterogeneity within microbial biofilms. InMicrobial Sediments. Riding, R.E., and Awramik, S.M. (eds). Heidelberg, Germany: Springer, pp. 9–15. Here, the authors propose that EPS limits salt mobilization and crystalization at the surface. Indeed, other rock weathering studies invoke the water retention capabilities of EPS as a way of maintaining acids and chelating agents in contact with the weathering surface. I appreciate that this could be less relevant under arid conditions, but again, the authors should explicitly say this and test their hypothesis that the water retention retards weathering experimentally.

- Table 1: This table doesn't seem like essential information and I suggest that the authors might instead place it in the Supplemental Information.

- The amplicon results figure in the supplement (Figure S2) is well done, and shows all of the data in a presentable manner. Personally, I would like to see this as a figure in the main body of the manuscript, rather than as a supplemental figure. Perhaps its position (supplement vs main) could be swapped for the current Figure 4 panels C and D?

---

## Author Comment (AC1) · 4 Jan 2019

[]article lmodern amssymb,amsmath ifxetex,ifluatex upquote []microtype [protrusion]basicmath hyphensurl [unicode=true]hyperref parskip

Dear Editor,

Please find attached a revised version of the manuscript, titled "The origin and role of biological rock crusts in rocky desert weathering". We thank you and the anonymous reviewer for the constructive comments and appreciate the time you have invested in improving this manuscript. The changes to the manuscript include text changes that address the points raised by the reviewer. Thank you for the efficient review process. We will be glad to answer any further questions.

Sincerely,

Roey Angel on behalf of all co-authors

**Comments by reviewer:**

**R: The origin and role of biological rock crust in rocky desert weathering" by Wieler et al. is focused on the origin and role of rock biofilms in cavernous weathering in arid and hyperarid climate. Authors use multiple techniques to reveal the origin of biocrust and its effect on evaporation rate and thus weathering. Manuscript contains valuable information and its worth of publication.**

**The answer to questions 1-15 in review instruction is positive, except the critical**

**comments mentioned below. Please take into account that I am not expert on DNA techniques nor on statistical processing of such data, so I can not reliably review chapters 2.6, 2.7 and 3.3 from biological point of view in required depth. These chapters seems to be however clear and makes sense to person from other scientific branch.**

A: Thank you, we appreciate this feedback.

**R: Page 1 line 12: It is unclear which portions of honeycombs or tafoni surfaces were sampled for biogenic rock crust (BRC). Was it the outer surfaces or hollows (cavities)?**

**R: Page 2 line 5: It seems that just outer surfaces are covered by BRC, but it is not clearly stated. It should be spelled our more clearly if BRC is missing in caverns or if it covers whole surface of tafoni.**

A: Thank you for bringing this up. We sampled the outer surfaces of the cavities. The BRC is missing in the caverns since the caverns are the weathering fronts. The text was corrected in page 1 line 12: "We studied the origin and role of rock biofilms covering rock surfaces in geomorphic processes of limestone and dolomitic rocks that feature comparable weathering morphologies though originating from arid and hyperarid environments, respectively". Correction was also done at page 2 lines 4-6: "Following the cementation processes, typical honeycomb features are formed on the exposed parent

rock, typified by pits separated by thin walls that are coated by the calcrete or dolocrete and are absent in pits cavities".

**R:Page 1 line 32: Reference at the end of sentence is needed**

A: Thank you, a reference for the ventifacts exists at line 33.

**R: Page 2 line 26**: **Fungi and algae are reported as common constituent of BRC by Slavik et al (2017)-cited in document**.

A: The reference was added to the manuscript.

**R: Page 3 line 1**: **There should be few more sentences given on characterization of limestone and dolomite: sedimentation settings, diagenesis, lithology, whether these rocks act as aquifer or aquitard, into which degree the water from rain infiltrates to them vs. surface runoff dominates**

A: Thank you, we presented detailed description of the subjected lithologies at the supplementary information (Table S1).

**R: Page 3 line 4: Rather than P/PET 0.05-0.005 you should write this ratio for both studied localities respectively (to show the difference between them). This ratio is in one of supplementary tables, but it should be also directly in the text.**

A: Correction was made in the manuscript in page 3 lines 3-4: "The Negev Desert, Israel, maintains arid to hyperarid conditions since the Holocene and has an aridity index (P/PET) of 0.05 for the arid region and 0.005 for the hyper-arid region (Amit et al., 2010; Bruins, 2012)".

**R: Page 3 line 10**: **These samples were taken from 1) narrow walls of tafoni, 2) hollows of tafoni, 3) outer surfaces, which are not covered by tafoni or 4) inner material below tafoni hollows? This should be clear. Similarly for each method used is important which of these four types of material you used.**

A: All rock samples were taken from outer surfaces that cover cavernous features.

Correction was made in the manuscript in page 3 line 9: "Twenty-four rock surface samples were collected along rocky slopes facing northward, comprising".

**R: Page 3 line 29**: **You mention measuring of porosity in direction normal and perpendicular to bedding. This is good idea, but please report also results from both measurements (table 1). Currently the direction is not distinguished there. Measured samples were without crust, with crust or crust itself? It should be more clearly spelled out in this (and also other) method(s), whether the underlying rock or crust was tested! If crust was not measured it will be valuable to measure the crust as well and compare it to underlying rock.**

A: Thank you for bringing this up. The porosity was measured on rock cylinders without crust, correction was made in the text in page 3 lines 29-30: "twelve rock core cylinder samples that their BRC was mechanically removed using a diamond saw (Dremel, Racine, WI, USA) to a depth of 5 cm". Crust porosity was not measured due to technical limitations. The porosity values of the normal and perpendicular directions showed no bigger difference, therefore we unified their results.

**R: Page 3 line 30**: **It is generally recommended to do about 20 readings by Schmidt hammer per single obtained value. Your 20 measurements per lithology means 20 readings (1 value) or 20 sites measured each by ?20? readings? Please specify. Also in further text you use "elasticity" (P5 L23), "surface penetration resistance"(Table 1). This cannot be measured by Schmidt hammer, but it could be possibly derived by some formula. Did you measure it by other device? (Please characterize the device) or did you calculated that from rebound value of Schmidt hammer (then please provide the formula and reference). The terms "penetration resistance" elasticity of material should be unifi̧ed in text and tables, if they describe the same measured parameter.**

A: The Schmidt hammer measurements contained 20 readings per single site, correction in the text is in page 3 line 34: "Twenty measurements were carried out for each

site bearing different lithology".

The values mentioned are the elastic rebound (R) measured by the Schmidt hammer. The terms were unified in the text as elastic rebound and corrected in the text in page 5 line 25-27: "To study the possible differences between these sites, we performed geological characterisation of 10 limestone and dolomite rocks collected from the arid and hyperarid sites, respectively, testing for mineral content, porosity, permeability and elastic rebound" and in table 1.

**R: Page 4 line 4 and 6: really it was delta18O in H2O? I would expect O in CO2 gas not H2O vapor. The sample is carbonate not water. Please check**

**R: Page 4 line 7 and 8: Really SMOW was used? All values seems to be referenced to PDB standard to me (Fig. 2). Please check.**

A: Thank you, we checked it again, and corrected the in page 4 lines 6-11: " Measurements (in duplicate) of $\delta^{18}$O-calcite and $\delta^{13}$C-DIC were performed on gas source isotope ratio mass spectrometer (GS-IRMS; Thermo Fisher Scientific, Waltham, MA, USA) coupled to a Gas Bench II interface (Thermo) after CO2 equilibration or CO2 extraction by acidification for $\delta^{18}$O-calcite and $\delta^{13}$C-DIC, respectively. The samples were calibrated against internal laboratory standards: carbonate standard NBS19. $\delta^{13}$C values were referenced relative to Vienna PeeDee Belemnite (VPDB) standard as previously described (Uemura et al., 2016) with SD of 0.1‰All values are reported in per-mil (‰".

**R: Page 4 line 17: What do you mean by "incubated"? bacteria were introduced to rock?**

A: The Desiccation experiment included rock cylinders with and without BRC. No bacteria were introduced. The cylinders were simply dried at 44 °C (low temperature was used to protect the BRC). The term "incubated" is indeed misleading and was removed.

**R: Page 5 line 16: "crusts were restricted to atmospherically exposed..." please**

**change "atmospherically exposed" to more proper description. Do you mean that crust was missing in bottom of hollows? Or deeper below surface? This should be more clear.**

A: Correction was made in the text in page 5 lines 18-19": The weathering and presence of the crusts were restricted to the upper parts of the rocks (i.e., rock parts that are exposed to the atmosphere)".

**R: Page 5 line 17 vs line 1**9: **Statement is not consistent. In first sentence you speak about weathering MORPHOLOGY in second you speak about weathering RATES. So if morphology is the same, this does not necessarily mean their rates are the same as well. I am afraid that this if fact not challenges the model.**

A: Thank you for pointing this out. The manuscript aims to explain the presence of the same weathering morphology across different lithologies and different climates in comparison to common model that suggest different parameters which may affect weathering rates that leads to weathering morphology. The sentence was changed to "The presence of an identical weathering morphology and its prevalence in different climates and lithologies challenges the current model, which assumes that surface permeability, moisture and the presence of salts as primary factors control the weathering type and rates" in order to resolve this inconsistency.

**R: Page 5 line 29-32: Text is unclear, please rewrite.**

A: In lines 29-32 we note that the honeycomb weathering features results from the protective effect by the presence of the BRC and not from different mineral composition, as was previously suggested. The text was corrected in page 5 lines 34-39: "Presence of thin septa between the weathering pits was previously suggested to result of different mineralised networks or case hardening (McBride and Picard, 2004), however, detecting calcretes and dolocretes on limestone and dolomite, respectively, on the rocks' surface reject this hypothesis.. In fact, the detection of mature calcretes and dolocretes could serve as an indication of atmospheric exposure but was also suggested to result

from biogenic activity (Alonso-Zarza and Wright, 2010; Goudie, 1996)."

**R: Page 6 line 35: "clogging the pores on the surface of the rock and thereby minimizing capillary rise". This statement is confusing. In fact the smaller the pores, the higher the capillary rise. The reason why biota affects capillary water is not the diminishing the size of pores but the presence hydrophobic organic matter. So please mention rather hydrophobicity here as explanation.**

A: We mention clogging effect as a result of the EPS originated by the biota. The desiccation experiment along with the different characteristics of the BRC note that small concentrations of EPS in the BRC are suggested to have a strong retarding effect on rock pores, thus effecting water movement through the rock. As a result the BRC mitigate crystallization of dissolved salts.

**R: Page 6 line 42: Slavik et al 2017 reports DECREASE of hydraulic conductivity and capillary water absorption by 15-300 times and 2-33 times, respectively. So cited statement that BRC does NOT affects water transport rates is wrong. Only diffusion of water vapor was not effected by crust based on Slavik et al 2017. But in your case it could be the same situation: lowered evaporation is not necessarily due to low diffusion of vapor (only few if any well designed studies indicated that biocrust is capable even to affect vapor transport). Far more probably the decreased evaporation rate (which you observe on crust) is due to fact that capillary front is pushed below surface due to hydrophobic organic matter and thus diffusion occurs via more thick dry surface layer in case of BRC sample compare to bare rock core (longer diffusion path means far lower evaporation rate). Until the vapor diffusion is measured via BRC and bare rock and both rates are compared on your samples (e.g. by wet cup technique) it is impossible to say if evaporation rate is lowered by(i)lower diffusion rate or(ii)due to different geometry of capillary front. This will be very valuable to test.**

A: Thank you for mentioning this, the text was corrected at page 7 lines 4-5: "Moreover,

the obtained results performed on temperate sandy stones showed decrease effect of BRC on water transport rates (Slavík et al., 2017)."

**R: Page 7 line 13: From which depth the samples were taken?**

A: Soil samples were collected beneath the soil crust, at a depth of 5 centimeter. Correction was made at page 3 line 13: "Concomitantly, six soil samples (ca. 500 g each) were collected at a depth of 5 cm beneath surface, half from the arid (named: SBSoil 1-3) and a half from the hyperarid (named: UVSoil 1-3)site".

**R: Page 8 line 28 and 29: Sentence not clear. "...composition and function." of what?**

A: Correction was made in the text in page 8 lines 28- 33: "The two BRCs did differ in their bacterial communities at the OTU and higher taxonomical levels, demonstrating a discrepancy between microbial communities composition and function".

**R: Page 9 line 4: please specify which "microscale conditions"**

A: Correction was made in the text in page 9 line 10: "The results presented here suggest that in arid environments, microscale climatic conditions determine the magnitude of weathering that shape the landscape".

Technical corrections:

**R: Page 1 line 13: replace "hard lime" by "limestone"**

A: Correction was made in page 1 line 13: "We studied the origin and role of rock biofilms covering rock surfaces in geomorphic processes of limestone and dolomitic rocks".

**R: Page 3 line 12: if UVSoil had 3 samples, there should not be "UVSoil 1-12" but rather"1-3"**

A: Correction was made in page 3 line 14: "half from the arid (named: SBSoil 1-3) and

a half from the hyperarid (named: UVSoil 1-3)".

**R: Page 4 line 10: Use rather "Evaporation experiment" then "Desiccation experiment"**

A: We used the term desiccation because the term evaporation does not include heating from an outer source as was conducted in this experiment.

**R: Page 5 line 12: replace "weathering features" by "Cavernous weathering features" to be clear which weathering features you mean**

A: We specify the cavernous weathering features out of many other weathering features found in rocky desert outcrops.

**R: Page 6 line 8: please replace 2.5 Ga by 2.5 Ma**

A: Correction was made in page 6 line 11:" The low ratio detected here (Fig 2A) and by Vaks et al. (2010) suggest that the Negev region has been able to support only limited vegetation for at least 2.5 Ma".

**R: Page 6 line 14: there should be rather "In" then" between". The sentence is not much clear (it is unclear if values are concerning BRC, underlying rock, or both; but clearly not the boundary between them - Fig 2).**

A: Thank you, the text was corrected at page 6 line21.

**R: Fig. 2: please add PDB standard to horizontal axis of fig. a**

A: Thank you, the figure was corrected.

**R: Fig. 3: please replace "desiccation" by "drying"**

A: Thank you, the figure and the text was corrected.

**Supplement:**

[revised manuscript text omitted]
 upper parts of the rocks (i.e., rock parts that are exposed to the atmosphere). The presence of an identical weathering morphology and its prevalence            20
in different climates and lithologies challenges the current model, which assumes that surface permeability, moisture and the presence of salts as primary factors control the weathering type and rate (Goudie et al., 2002; Smith, 1988; Smith et al., 2005). However, the lack of correlation between tafoni weathering magnitude and climate has already been reported (Brandmeier et al., 2011).

To study the possible differences between these sites, we performed geological characterisation of 10 limestone and            25
dolomite rocks collected from the arid and hyper-arid sites, respectively, testing for mineral content, porosity, permeability and elastic rebound. As expected, our results showed different lithological parameters between the limestone and dolomite rocks (Table 1), yet they displayed similar weathering features. Moreover, petrographic thin section analysis showed that on both rock types, crusts had developed to a similar thickness of 1-6 mm, irrespective of climatic conditions including mean annual precipitation (Fig. 1C). Nevertheless, microclimatic conditions, like dew or surface temperature may still            30
impact local morphologies. In addition, the thin section analysis showed that the crusts are composed of masses of micritic to microsparitic minerals that form laminated structure (Fig. 1C). Such laminated structures indicate that the crusts are stage four terrestrial calcretes and dolocretes, suggesting a mature crust phase (Alonso-Zarza and Wright, 2010).  Presence of thin septa between the weathering pits was previously suggested to result  of different mineralised networks or case hardening (McBride and Picard, 2004), however, detecting calcretes and dolocretes on limestone and dolomite,            35
respectively, on the rocks' surface reject this hypothesis.. In fact, the detection of mature calcretes and dolocretes 
[revised manuscript text omitted]

---

## Author Comment (AC2) · 6 Feb 2019

[]article lmodern amssymb,amsmath ifxetex,ifluatex upquote []microtype [protrusion]basicmath hyphensurl [unicode=true]hyperref parskip

Dear Editor,

Please find attached a revised version of the manuscript, titled "The origin and role of biological rock crusts in rocky desert weathering". I thank you and the anonymous reviewer for the constructive comments and appreciate the time you have invested in improving this manuscript. The changes to the manuscript include text changes that address the points raised by the reviewer. Thank you for the efficient review process. I will be glad to answer any further questions.

[Figure]

Sincerely,

Roey Angel on behalf of all Co-Authors

**Comments by reviewer:**

**R: In the manuscript " The origin and role of biological rock crust in rocky desert weathering" by Wieler et al., the authors set out to characterize the microbial communities associated with rock crusts on limestone and dolomite host rocks sampled from arid regions. In this aspect they have succeeded. The authors also claim to have discovered how crust-associated microbial communities influence the mediation of weathering processes associated with these clasts. With respect to this second claim, the authors have only shown here that EPS associated with the microbial communities helps the rock surface to retain water, not that the water retention mitigates the weathering process via slowing crystal growth (as is claimed). The finding of EPS retaining water has been shown before in other environments (refs below), but in those studies, the retention of water was proposed to enhance weathering via various mechanisms, not retard it. Perhaps if experiments showed that rock weathering decreased under EPS free portions of the rock surface, I would find this second claim convincing, but these experiments/data are not present. I suggest that the authors rework the manuscript to focus only on the characterization of the community, and not on biogenicity aspects of the crust formation that are not supported by the research findings.**

A: Thank you for comment, we appreciate this feedback and have addressed all the specific comments below. We specifically addressed the issue of the influence of biological crusts on rock weathering processes in detail below.

**R: The authors frequently misspell words that should contain the letter "z" but instead are spelled with an "s" (stabilise vs stabilize; colonise vs. colonize etc.). Perhaps this is a US vs British spelling difference, but the journal editors may want to clarify which style they want used.**

A: Indeed, these words are spelled with "s" under the British spelling system. We

followed the British spelling system throughout the manuscript as is customary for European journals.

**R: Page 2, Line 14, This sentence could be rewritten for clarity.**

A: The sentence was rewritten in page 2 lines 12-14: "Recently, Bruthans and colleagues (2018) conclusively demonstrated that in temperate climate moisture flux followed by salt crystallisation at the boundary layer govern the case hardening model."

**R: Page 5 Line 12, please provide the number of samples that contain weathering features.**

A: Ten rock samples, from each lithology, were studied and characterized. The number of samples is mentioned in page 5 lines 21-23: "To study the possible differences between these sites, we performed geological characterisation of 10 limestone and dolomite rocks collected from the arid and hyperarid sites, respectively, testing for mineral content, porosity, permeability and elasticity".

**R: Page 5 Line 38, I disagree with the authors' use of the terms "biogenic" to refer to the rock crust. Let's assume that the 13C depleted values results from the liberation of carbon from photosynthetic materials via respiration (there are other ways to get 13C depleted carbonate, but let's just assume the mechanism the authors invoke is correct), that CO2 should then be creating an acidic environment that does not necessarily favor carbonate formation. More importantly, a carbon contribution from respiration recorded in a carbonate does not make a rock crust any more "biogenic" than any carbonate that forms in any environment in which CO2 is sourced from respiration, which could be any environment! I strongly recommend that the authors remove biogenic from these paragraphs, as the carbonate carbon isotope data do not demonstrate that living processes were necessary (or even important) for the carbonate crust formation.**

A: We did not measure carbonate rather we measured the isotopic signature of the

all carbon form together using pyroslysis. The negative values found in the crust layer indicate a mixture of marine carbonate sedimentation (with d13C values close to 0 per mill) with freshly photosynthesised (not respired!) carbon (which typically has a d13C values of -20 – -30 per mill for photosynthetic microorganisms). Since no other terrestrial process, but photosynthesis, is known to generate such low 13C values such values are considered a very reliable signature of biological carbon fixation. The term "biogenic" was therefore used in this context to indicate the contribution of a biological process to the carbon pool and to differentiate it from the parent rock material, which had a clear abiotic signature of marine carbonates. We further do not suggest that the crust was formed only by direct precipitation of microbial activity but it is a mixture of both organic and inorganic materials that binds together. This is in fact what defines biological soil and rock crusts.

**R: Page 6, Line 28, please provide the full citation information for the Jiang paper**

A: The citation was corrected in the text in page 6 line 28: Jiang et al., 2004, and in the reference list-"Jiang, W., Saxena, A., Song, B., Ward, B. B., Beveridge, T. J. and Myneni, S. C. B.: Elucidation of Functional Groups on Gram-Positive and Gram-Negative Bacterial Surfaces Using Infrared Spectroscopy, Langmuir, 20, 11433-11442, 2004".

**R: Page 6, Line 29, the spatial correlation of a biofilm with a mineral precipitate DOES NOT establish that the biofilm was involved in the formation of the mineral formation. As an example, the modern day La Brea tar pits contain abundant bacteria and archaea, that does not mean that those bacteria and archaea are responsible in any way for the presence of the tar of the fossils embedded in the tar, despite their spatial correlation. The same is true for our teeth, or for certain modern stromatolites. I'm certainly not saying that there aren't many cases where microbes are involved in mineral precipitation, there clearly are many, included microbes involved in carbonate formation, but in cases like this, it can be difficult to demonstrate this relationship and we should be careful with our**

**words and our claims.**

A: We suggest that the laminated rock crust results from microbial activity. These laminated fabrics resembled to similar fabrics in marine stromatolites. Such fabrics in marine environments were linked to diverse sediment-microbes interactions and included diverse microbial communities (Bosak, Liang, Sim, & Petroff, 2009; Dupraz et al., 2009).The mediation of microbial activity in the crust formation, in our study, is mainly suggested as a binding agent to form thin coatings on the rock surface. We support this statement with the presence of the EPS and the isotope measurements that were found in the rock crust and were absent in the host rock. We further do not suggest that the crust was formed only by direct precipitation of microbial activity but it is a mixture of both organic and inorganic materials that binds together.

**R: Page7, Line 5, the description of the observed vs. predicted phylotypes (predicted by Chao/ACE) is unclear and should be better described.**

A: The text was corrected in page 7 line 5: "The communities of the BRC showed an average of 182 observed, 354 predicted bacterial phylotypes, and Shannon's H was 3.8 (Fig. 4A; Table S2), for arid limestone and 129 observed, 315 predicted phylotypes and Shannon's H was 3.3 for hyperarid dolomite, with no significant difference between the rock types."

**R: Page 8, "specialism" should be "specialization"**

A: The text was corrected in page 8 line 32: "The BRC communities also differed from their surrounding soil and dust, indicative of the specialization of the colonising taxa to rock environment."

**Page 9, Line 9 says "were was", but should be just "was"**

A: The text was corrected in page 7 line 9: "The diversity of the dust samples was as poor as the BRC's (169 and 107 observed and predicted OTUs and Shannon's H = 3.0 and 1.5, on average) and did not differ between sites (Fig. 4A; Table S2)."

**Page 9, Line 17 – this sentence could be rewritten for clarity**

**Page9, Line18–This sentence should probably be ended with "respectively)" to indicate which percentages with each parameter, or better yet rewrite the entire paragraph and give a sentence to each parameter.**

A: The text was corrected in page 7 line 17-18: "Beta-diversity analysis, using variance partitioning, showed statistically significant differences between samples on the OTU-level based on climate, sample type (i.e. rock, soil or dust), and to a small extent also via their interaction. These variables were found to significantly contribute to the differences in bacterial communities accounting for 22%, 40% and 3.8% of the total variance, respectively (Fig. 4B, Table S3)."

**Figure 4b – the dust samples are hard to differentiate using the current color.**

A: We increased the colour intensity of the dust samples for better clarity.

**I understand that the rock crusts studied here are not the same as the manganese oxidize-rich rock varnish that has been extensively studied elsewhere, but are the microbial communities similar or different? Would this be worth mentioning as a point of comparison? Some readers will be more familiar with those features.**

A: A comparison between the microbial communities of the manganese oxidize rock varnish and the ones in the rock crusts mentioned briefly in our work in page 2 lines 26-27, page 7 lines-13,16,26 and page 8 line 3. The comparison refers to the work conducted by Lang Yona et al., 2018.

**The authors propose that the microbial community should be similar to that of the surrounding soil, or incoming dust, if those are the sources, but then demonstrate with their amplicon results, that the communities on the rocks are substantially different from those in the soil and dust. This is an interesting result and worthy of publication for its own sake in my view. I think the authors do a nice**
**job with this part of the paper and should be commended.**

A: Thank you, we appreciate this feedback.

**The presented results do appear to show that the biofilm contributes to the retention of water at the rock surface. However, this is not a new claim and there are numerous other papers in the older literature that also show this (Potts, M. (1999) Mechanisms of desiccation tolerance in cyanobacteria. Eur J Phycol 34: 319–328.; Decho, A.W. (2000) Exopolymer microdomains as a structuring agent for heterogeneity within microbial biofilms. In Microbial Sediments. Riding, R.E., and Awramik, S.M. (eds). Heidelberg, Germany: Springer, pp. 9–15. Here, the authors propose that EPS limits salt mobilization and crystalization at the surface. Indeed, other rock weathering studies invoke the water retention capabilities of EPS as a way of maintaining acids and chelating agents in contact with the weathering surface. I appreciate that this could be less relevant under arid conditions, but again, the authors should explicitly say this and test their hypothesis that the water retention retards weathering experimentally.**

A: The current knowledge dealing with retention water by EPS, as mentioned, has limited data on the role they might play in correlation with arid rock morphologies. Cavernous weathering features result from three initiation conditions: porous media, salt solution and hydration-desiccation cycles (Scherer, 1999, 2000). We followed the current knowledge on EPS, found in the rock crust, to test how its water retention abilities, may correlate to arid rock morphology. To do so, we ran a desiccation experiment, discussed in the manuscript. This experiment showed that rock crusts, containing EPS, retard water at the rock surface. Resulting from this experiment we note that the rock crust containing EPS limits the water needed for generating salt solution at the rock interface, and limits rock weathering over small spatial scales .

**Table 1: This table doesn't seem like essential information and I suggest that the authors might instead place it in the Supplemental Information.**

A: The table was moved to the supplementary

**The amplicon results figure in the supplement (Figure S2) is well done, and shows all of the data in a presentable manner. Personally, I would like to see this as a figure in the main body of the manuscript, rather than as a supplemental figure. Perhaps its position (supplement vs main) could be swapped for the current Figure 4 panels C and D?**

A: We now include Figure S2 as part of the main text. However we kept panels C and D in Figure 4 because they illustrate the real proportions of various taxa rather than a difference in a binary comparison

**Supplement:**

[revised manuscript text omitted]
 Mainzar E C Mozdzar T I Datsch S Datt Didge I Progitzer K S Daymond D A Diabe C S Shumeker K                                                                                     |     |
| System Crier, A. Welter, D. and Vee, K. Twelve testelle hypotheses on the cochigle of sphering. Cochigle of $0(2)$                                                                 |     |
| Sutton-Orier, A., waiter, K. and Yoo, K.: Twelve testable hypotheses on the geoblology of weathering, Geoblology, 9(2),                                                            |     |
| 140-165, doi:10.1111/j.1472-4669.2010.00264.x, 2011.                                                                                                                               | 1.5 |

[revised manuscript text omitted]
       | <del>Quartz</del> | Dolomite                                | Calcium | Quartz |
| BRC mineralogy (%)                                    | 0                                  | 95     | 3                 | <del>90</del>                           | 2       | 1      |
| Host rock mineralogy (%)                              | 0                                  | <del>95</del> | 0                 | <del>95</del>                           | 0       | 4      |
| Porosity (%)                                          | $\frac{13.5 \pm 2.2}{2.2}$         |               |                   | <del>8.25 ± 1.3</del>                   |         |        |
| Permeability (miliDarcy)                              | <del>0.1 – 3.8</del>               |               |                   | <del>0.05 – 0.41</del>                  |         |        |
| Surface penetration resistance (kg cm -1 ) | <del>100 – 365</del>               |               |                   | <del>130–230</del>                      |         |        |

---

## Author Response (AR1)

Dear Editor,

Please find attached a revised version of the manuscript, titled "The origin and role of biological rock crusts in rocky desert weathering". I thank you and the anonymous reviewers for the constructive comments and appreciate the time you have invested in improving this manuscript. The changes to the manuscript include text changes that you addressed. Thank you for the efficient review process. I will be glad to answer any further questions.

Sincerely,

Roey Angel on behalf of all Co-Authors

**R: The dust and rock samples were collected a full year apart. How can you be sure that the communities are representative of the conditions at both time points? Meaning, how do you know that the differences between dust, soil and rock are not due to temporal changes? Acknowledging this experimental fact would be valuable on Pg. 8.**

A: Soil and rock microbial communities in arid regions result from a long term process (decades), and typically exhibit rapid changes after hydration events but little to no seasonal changes (see e.g. Angel and Conrad 2013, Barnard et al. 2013, Šťovíček et al. 2017). We therefore assumed in this study that the temporal aspect is minor (on the seasonsal-yearly term, or course). The microbial community in the settled dust was tested as a potential source, like the soil, however major dust storms that can be sampled occur only at specific time in the year which limits the sampling campaign. This is now mentioned in Pg. 3 and 7.

**R: Be consistent with spelling of hyperarid. In some cases you write hyper-arid and others hyperarid.**

A: The text was corrected, we now use 'hyperarid' throughout the manuscript.

**R: P. 1, L. 33: replace [ with a comma**

A: The text was corrected.

**R: P. 3, L. 16: How many replicate dust samples?**

A: We collected two dust samples from each site where each sample was a composite of two sub-samples. The text was corrected.

**R: P. 3, l. 29-30: revise to "samples that had their BRC mechanically removed using a diamond saw (Dremel, Racine, WI, USA) to a depth of 5 cm,"**

A: The text was corrected: "Six samples were taken from each lithology, that their BRC was mechanically removed using a diamond saw (Dremel, Racine, WI, USA) to a depth of 5 cm, each set of six samples were prepared in two orthogonal directions providing the normal to bedding and parallel to bedding."

**R: P. 4, l. 3-5: no spaces between delta and the isotope.**

A: The text was corrected.

**R: P. 4, l. 15-20: this isn't clear. Can you provide more info on how the cores were covered with epoxy (Devcon) and aluminium foil? Its not clear how "only the upper base of the crusted and bare cylinders" were left uncovered. Also, correct the spelling of aluminum.**

A: The text was revised for clarity: "Each cylinder was immersed in distilled water for 72 h. Then, the entire surface of the cylinder, except the upper base was , covered with epoxy (Devcon) and then with aluminium foil, leaving only the upper basetop of the crusted and bare cylinders (with or without BRC) uncovered to allow evaporation." Spelling of the word aluminium was corrected.

**R: P. 5, L. 27: 16S rRNA is not italicized**

A: The text was corrected.

**R: P. 5, L. 20: change to Illumina MiSeq**
A: The text was corrected.

**R: P. 5, l. 34: please consider putting all of your code for sequence processing and R analysis in the supplemental material. This is not required but it's a great way of making your data more accessible for the reader and for future metanalysis. If you don't opt this time please consider in future papers.**
We also believe in sharing data and analysis code enhancing reproducibility and supporting meta-analysis. We already uploaded the scripts and datasets to GitHub, but embarrassingly enough forgot to mention this in the manuscript. Thank you very much for pointing this out! This now appears in Pg. 9 in a "Data availability" section at the end of the manuscript, as required.

**R: Pg. 6, l. 44: change to drying experiment**
A: The text was corrected.

**R: Pg. 7, l. 3-4: revise to "inevitably lead to decreased weathering rates."**
A: The text was corrected.

**R: Pg. 7, l. 4-5: This sentence is awkward, revise for clarity. Maybe, "Moreover, our results differ from thos performed on temperate sandy stones that showed no effect of BRC on decreasing water transport rates (Slavík et al., 2017)."**
A: The text was revised.

**R: Pg. 7, l. 8: omit "a"**
A: The text was corrected.

**R: Pg. 7, l. 15: change wording –first there is were and was and it should be was. Second, I wouldn't call diversity "poor"—low is a better term**
A: "poor" was contrasted with "rich" in the context of biodiversity and differs from high/low diversity. Since we measured both richness and diversity and both were found to be low, we tried to distinguish the two. However, for better clarity we now modified the sentence to: "As expected, we found simple BRC communities (i.e. low-richness and low-diversity). "

**R: Pg. 8, l. 9: this doesn't make sense: "The BRC bacterial communities were previously described". You mean other BRC communities have been described not the same ones you are working on. Revise to be more clear that you are referring to other data and how your work is different.**
A: Yes, we meant "other BRC communities". The text was corrected.

**R: Pg. 8, l. 12: "the two BRCs we studied"**
A: The text was corrected.

**R: Pg. 8, l. 35-37: not clear what you mean by a discrepancy between structure and function. Function is really only potential function in this case since you are doing DNA-based work**

A: The text was revised. 'Structure' refers to community structure while 'function' refers to building a crust.

[revised manuscript text omitted]